# NEIGHBOR-ENCODER

## ABSTRACT

We propose a novel unsupervised representation learning framework called *neighbor-encoder* in which domain knowledge can be trivially incorporated into the learning process without modifying the general encoder-decoder architecture. In contrast to autoencoder, which reconstructs the input *data*, neighbor-encoder reconstructs the input data's *neighbors*. The proposed neighbor-encoder can be considered as a generalization of autoencoder as the input data can be treated as the nearest neighbor of itself with zero distance. By reformulating the representation learning problem as a neighbor reconstruction problem, domain knowledge can be easily incorporated with appropriate definition of similarity or distance between objects. As such, any existing similarity search algorithms can be easily integrated into our framework. Applications of other algorithms (e.g., association rule mining) in our framework is also possible since the concept of "neighbor" is an abstraction which can be appropriately defined differently in different contexts. We have demonstrated the effectiveness of our framework in various domains, including images, time series, music, etc., with various neighbor definitions. Experimental results show that neighbor-encoder outperforms autoencoder in most scenarios we considered.

## 1 INTRODUCTION

Unsupervised representation learning has been shown effective in tasks such as dimension reduction, clustering, visualization, information retrieval, and semi-supervised learning (Goodfellow et al., 2016; Yang et al., 2017). While domain-specific unsupervised representation learning methods like word2vec (Mikolov et al., 2013a;b) and video-based representation learning (Agrawal et al., 2015; Jayaraman & Grauman, 2015; Wang & Gupta, 2015; Pathak et al., 2017) have been widely adopted in their respective domains, their success cannot be directly transferred to other domains as their assumptions do not hold for other types of data. In contrast, general unsupervised representation learning methods such as autoencoder (Bengio et al., 2007; Huang et al., 2007; Vincent et al., 2008; 2010) can be effortlessly applied to data from various domains, but the performance of general methods is usually inferior to those that utilizes domain knowledge (Mikolov et al., 2013a;b; Agrawal et al., 2015; Jayaraman & Grauman, 2015; Wang & Gupta, 2015; Pathak et al., 2017).

In this work, we propose an unsupervised representation learning framework (i.e., neighbor-encoder) which is *general* as it can be applied to various types of data and *versatile* since domain knowledge can be trivially added by adopting various "off-the-shelf" data mining algorithms for finding neighbors. Figure 1 previews the $t$-Distributed Stochastic Neighbor Embedding ($t$-SNE) (Maaten & Hinton, 2008) visualization produced from a human physical activity data set (see Section 4.3 for details). The embedding is generated by projecting representation learned by neighbor-encoder, representation learned by autoencoder, and raw data respectively to $2D$. By using a suitable neighbor finding algorithm, the representation learned by neighbor-encoder provides a more meaningful visualization than its rival methods.

In summary, our major contributions include:

- We propose a general and versatile framework, the neighbor-encoder, which can be used to trivially combine a large family of similarity search techniques with unsupervised representation learning to incorporate domain knowledge.
- We demonstrate the superior performance of the representations learned by neighbor-encoder, compared to representations learned by autoencoder in handwritten digit data,

image, human physical activity data, and instrumental sound data for various machine learning tasks including classification, clustering, and visualization.

- We demonstrate that the neighbor-encoder framework can considerably outperform autoencoder with an appropriate neighbor definition.

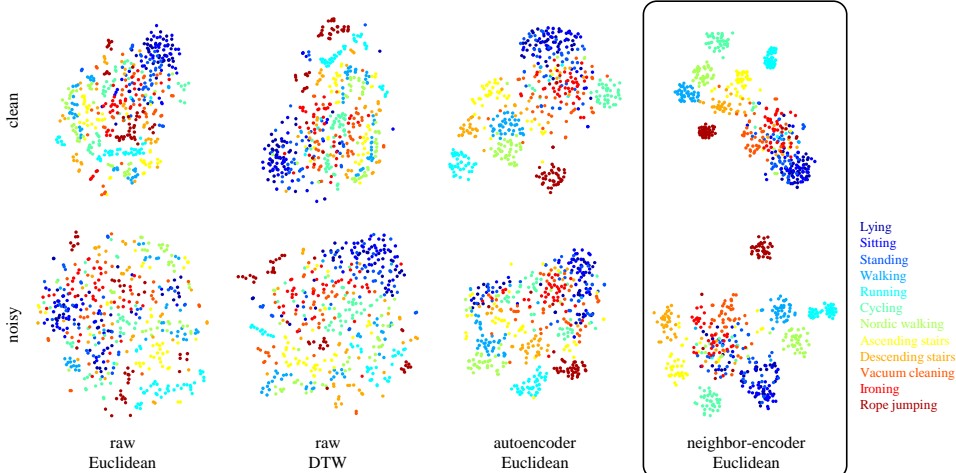

Figure 1: Visualizing the learned representation versus the raw time series on PAMAP2 (human physical activity) data set (Reiss & Stricker, 2012a;b) using t-SNE (Maaten & Hinton, 2008) with either Euclidean or dynamic time warping (DTW) distance (Nguyen et al., 2017). If we manually select 27 dimensions of the time series that are *clean* and relevant (acceleration, gyroscope, magnetometer, etc.), the representation learned by both autoencoder and neighbor-encoder achieves better class separation than raw data. However, if the data include *noisy* and/or irrelevant dimensions (heart rate, temperature, etc.), neighbor-encoder outperforms autoencoder noticeably.

## 2 RELATED WORK

**Unsupervised representation learning** is usually achieved by optimizing either domain specific objectives or general unsupervised objectives. For example, in the domain of computer vision and music processing, unsupervised representation learning can be formulated as a supervised learning problem with surrogate labels generated by exploiting the temporal coherence in videos and music (Agrawal et al., 2015; Jayaraman & Grauman, 2015; Wang & Gupta, 2015; Pathak et al., 2017; Huang et al., 2017); in the case of natural language processing, word embedding can be obtained by optimizing an objective function that "pushes" words occurring in a similar context (i.e., surrounded by similar words) closer in the embedding space (Mikolov et al., 2013a;b). Alternatively, general unsupervised objectives are also useful for unsupervised representation learning. For example, both autoencoder (Bengio et al., 2007; Huang et al., 2007; Vincent et al., 2008; 2010) and dictionary learning (Mairal et al., 2009) are based on minimizing the self-reconstruction error, while optimizing the $k$-means objective is shown effective in Coates & Ng (2012) and Yang et al. (2017). Other objectives, such as self-organizing map criteria (Kohonen, 1982; Bojanowski & Joulin, 2017) and adversarial training (Goodfellow et al., 2014; Donahue et al., 2016; Radford et al., 2015) are also demonstrated as effective objectives for unsupervised representation learning.

**Autoencoder** is a decade-old unsupervised learning framework for dimension reduction, representation learning, and deep hierarchical model pre-training; many variants have been proposed since its initial introduction (Bengio et al., 2007; Goodfellow et al., 2016). For example, the denoising autoencoder reconstructs the input data from its corrupted version; such modification improves the robustness of the learned representation (Vincent et al., 2010). Variational autoencoder (VAE) regularizes the learning process by imposing a standard normal prior over the latent variable (i.e., representation), and such constraints help the autoencoder learn a valid generative model (Kingma & Welling, 2013; Rezende et al., 2014). Makhzani et al. (2015) and Larsen et al. (2015) further improved generative model learning by combining VAE with adversarial training. Sparsity constraints

on the learned representation are another form of regularization for autoencoder to learn a more discriminating representation for classification, both the $k$-sparse autoencoder (Makhzani & Frey, 2013; 2015) and $k$-competitive autoencoder (Chen & Zaki, 2017) incorporate such ideas.

# 3 NEIGHBOR-ENCODER FRAMEWORK

In this section, the proposed neighbor-encoder framework is introduced and compared with autoencoder. Figure 2 shows different encoder-decoder configurations for both neighbor-encoder and autoencoder. In the following sections, we will discuss the motivation and design of each encoder-decoder configuration in detail.

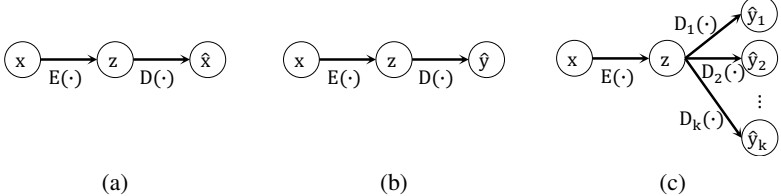

(a)  (b)  (c)

Figure 2: Various encoder-decoder configurations for training autoencoder and neighbor-encoder: a) autoencoder, b) neighbor-encoder, and c) $k$-neighbor-encoder with $k$ decoders.

## 3.1 AUTOENCODER

The overall architecture of autoencoder consists of two components: an *encoder* and a *decoder*. Given input data $x$, the encoder $E(\cdot)$ is a function that encodes $x$ into a latent representation $z$ (usually in a lower dimensional space), and the decoder $D(\cdot)$ is a function that decodes $z$ in order to reconstruct $x$. Figure 2a shows the feed-forward path of an autoencoder where $z = E(x)$ and $\hat{x} = D(z)$. We train the autoencoder by minimizing the difference between the input data $x$ and the reconstructed data $\hat{x}$. Formally, given a set of training data $X$, the parameters in $E(\cdot)$ and $D(\cdot)$ are learned by minimizing the objective function $\sum_{x \in X} loss(x, \hat{x})$ where $\hat{x} = D(E(x))$. The particular loss function we used in this work is cross entropy, but other loss function like mean square error or mean absolute error can also be applied. Once the autoencoder is learned, any given data can be projected to the latent representation space with $E(\cdot)$. Both the encoder and the decoder can adopt any existing neural network architecture such as multilayer perceptron (Bengio et al., 2007), convolutional net (Huang et al., 2007), or long short-term memory (Hochreiter & Schmidhuber, 1997; Srivastava et al., 2015).

## 3.2 NEIGHBOR-ENCODER

Similar to the autoencoder, neighbor-encoder also consists of an encoder and a decoder. Both the encoder and the decoder in neighbor-encoder work similarly as their counterpart in autoencoder; the major difference is in the objective function. Given input data $x$ and the neighborhood function $N(\cdot)$ (which returns the neighbor $y$ of $x$), the encoder $E(\cdot)$ is a function that encodes $x$ into a latent representation $z$ and the decoder $D(\cdot)$ is a function that reconstructs $x$'s neighbor $y$ by decoding $z$. Figure 2b shows the feed-forward path of a neighbor-encoder where $z = E(x)$ and $\hat{y} = D(z)$. Formally, given a set of training data $X$ and a neighborhood function $N(\cdot)$, the neighbor-encoder is learned by minimizing the objective function $\sum_{x \in X} loss(y, \hat{y})$, where $y = N(x)$ and $\hat{y} = D(E(x))$. Note that here "neighbor" can be defined in a variety of ways. We will introduce examples of different neighbor definitions later in Section 3.4.

We argue that neighbor-encoder can better retain the similarity between data samples in the latent representation space comparing to autoencoder. Figure 3 builds an intuition for this claim. As shown in Figure 3a, we assume the data set of interest consists of samples from two classes (i.e., blue class and red class, and each class forms a cluster) in $2D$ space. Since the autoencoder is trained by mapping each data point to itself, the learned representation for this data set would most likely be a rotated and/or re-scaled version of Figure 3a. In contrast, the neighbor-encoder (trained with nearest neighbor relation shown in Figure 3b) would learn a representation with much less

intra-class variation: as Figure 3c shows, when several similar data points share the same nearest neighbor, the objective function will force the network to generate exactly the same output for these similar data points, thus forcing their latent representation (which is the input of the decoder) to be very similar.

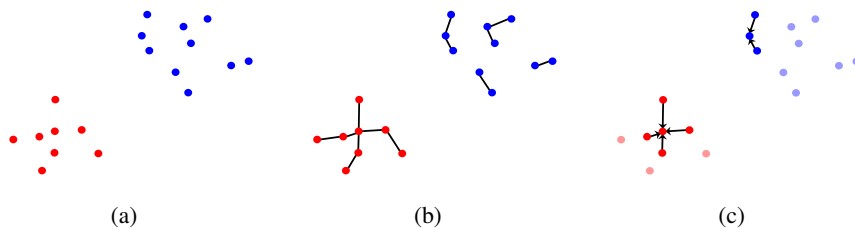

(a)                              (b)                              (c)

Figure 3: Intuition behind neighbor-encoder comparing to autoencoder. a) A simple $2D$ data set with two classes, b) the nearest neighbor graph constructed for the data set (arrowheads of the edges are removed for clarity), and c) an example of how neighbor-encoder would generate representation with smaller intra-class variation for highlighted data points. The neighbor-encoder learns similar representation for closely located data points by forcing these data points to reconstruct the same data point as these data points are most likely sharing the same nearest neighbor (shown by the arrows).

Since we are using neighbor finding algorithms to guide the representation learning process, one may argue that we could instead construct a graph using the neighbor finding algorithm, then apply various graph-based representation learning methods like the ones proposed in Perozzi et al. (2014), Tang et al. (2015), Grover & Leskovec (2016), Dong et al. (2017) or Ribeiro et al. (2017). Graph-based methods are indeed valid alternatives to neighbor-encoder; however, they have the following two limitations: 1) If one wishes to encode a newly obtained data, the out-of-sample problem would bring about additional complexity as these methods are not designed to handle such scenario. 2) It will be impossible to learn a generative model, as graph-based methods learn the representation by modeling the relationship between examples in a data set rather than modeling the example itself. As a result, the proposed neighbor-encoder is preferred over the graph-based methods when the above limitations are crucial.

### 3.3 $k$-NEIGHBOR-ENCODER

Similar to the idea of generalizing 1-nearest neighbor classifier to $k$-nearest neighbor classifier, neighbor-encoder can also be extended to $k$-neighbor-encoder by reconstructing $k$ neighbors of the input data (see Figure 2c). We train $k$ decoders which simultaneously reconstruct all $k$ neighbors of the input. Given an input data $x$ and the neighborhood function $N(\cdot)$ (which returns the $k$ neighbors $[y_i | \forall i \in \mathbb{Z} : 0 < i \leq k]$ of $x$), the encoder $E(\cdot)$ is a function that encodes $x$ into a latent representation $z$. Then, we have a set of $k$ decoders $[D_i(\cdot) | \forall i \in \mathbb{Z} : 0 < i \leq k]$, in which each individual function $D_i(\cdot)$ decodes $z$ in order to reconstruct $x$'s $i$th neighbor $y_i$.

The learning process of $k$-neighbor encoder is slightly more complicated than the neighbor-encoder (i.e., 1-neighbor-encoder). Given a set of training data $X$ and a neighborhood function $N(\cdot)$, the $k$-neighbor-encoder can be learned by minimizing $\sum_{x \in X} \sum_{y_i \in N(x)} loss(y_i, \hat{y}_i)$ where $\hat{y}_i = D_i(E(x))$ and $0 < i \leq k$. Note that since there are $k$ decoders, we need to assign each $y_i$ to one of the decoders. If there are "naturally" $k$ types of neighbors, we can train one decoder for each type of neighbor. Otherwise, one possible decoder assignment strategy is choosing the decoder that provides the lowest reconstruction loss for each $y_i \in N(x)$. This decoder assignment strategy would work if each training example has less than $k$ "modes" of neighbors.

### 3.4 NEIGHBORHOOD FUNCTION

To use any of the introduced neighbor-encoder configurations, we need to properly define the term neighbor. In this section, we discuss several possible neighborhood functions for the neighbor-encoder framework. Note that the functions listed in this section is just a small subset of all the available functions, to demonstrate the versatility of our approach.

- **Simple Neighbor** is defined as the several few objects that are closest to a given object in Euclidean distance or other distances, assuming the distance between every two objects is computable. For example, given a set of objects $[x_1, x_2, x_3, ..., x_n]$ where each object is a real-value vector, the neighboring relationship among the objects under Euclidean distance can be approximately identified by construing a $k$-$d$ tree.

- **Feature Space Neighbor** is very similar to *simple neighbor*, except that instead of computing the distance between objects in the space where the reconstruction is performed (e.g., the raw-data space), we compute the distance in an alternative representation or feature space. To give a more concrete example, suppose we have a set of objects $[x_1, x_2, x_3, ..., x_n]$ where each object is an audio clip in mel-frequency spectrum space. Instead of finding neighbors directly in the mel-frequency spectrum space, we transform the data into the Mel-frequency Cepstral Coefficient (MFCC) space as neighbors discovered in MFCC space is semantically more meaningful and searching in MFCC space is more efficient.

- **Spatial or Temporal Neighbor** defines the neighbor based on the spatial or temporal closeness of objects. Specifically, given a set of objects $[x_1, x_2, x_3, ..., x_n]$ where the subscript denotes the temporal (or spatial) arrival order, $x_i$ and $x_j$ are neighbors when $|i - j| < d$ where $d$ is a predefined size of the neighborhood. The skip-gram model in word2vec (Mikolov et al., 2013a;b) is an example of spatial neighbor-encoder, as the skip-gram model can be regarded as reconstructing the spatial neighbors (in a form of one-hot vector) of a given word.

- **Time Series Subspace Neighbor** is defined for multidimensional time series data as the similarity between two objects is measured by only a subset of all dimensions. By ignoring some dimensions, a time series could find neighbors with higher quality since it is very likely that some of the dimensions contain irreverent or noisy information (i.e., room temperature in human physical activity data) (Yeh et al., 2017). Given a multidimensional time series, we can use $m$STAMP (Yeh et al., 2017) to evaluate the neighboring relationship between all the subsequences within the time series.

- **Side Information Neighbor** defines the neighbor with side information which could be more semantic meaningful comparing to aforementioned functions. For example, images shown in the same eCommerce webpage (e.g., Amazon[1]) would most likely belong to the same merchandise, but they can reflect different angles, colors, etc., of the merchandise. If we select a random image from a webpage and assign it as the nearest neighbor for all the other images in the same page, we could train a representation that is invariant to view angles, lighting conditions, and product variations (e.g., different color of the same smart phone), etc. One may consider that using such side information implies a supervised learning system instead of an unsupervised learning system. However, note that the information provided to the system is still very limited: we only have the information regarding similar pairs while the information regarding dissimilar pairs (i.e., negative examples) are missing[2], and such limitation would hinder the performance of existing method like Siamese network (Koch et al., 2015).

## 4 EXPERIMENTAL EVALUATION

In this section, we show the effectiveness and versatility of neighbor-encoder compared to autoencoder by performing experiments on handwritten digits, images, human physical activities, and instrumental sounds data with different neighborhood functions. As the neighbor-encoder framework is a generalization of autoencoder, all the variants of autoencoder (e.g., denoising autoencoder (Vincent et al., 2010), variational autoencoder (Kingma & Welling, 2013; Rezende et al., 2014), $k$-sparse autoencoder (Makhzani & Frey, 2013; 2015), or adversarial autoencoder (Makhzani et al., 2015)) can be directly ported to the neighbor-encoder framework. As a result, here we did not exhaustively test all variants of autoencoder/neighbor-encoder, but only selected three most popular

---

[1] https://www.amazon.com/

[2] We can construct a 1-nearest-neighbor graph by treating each image as a node and connecting each image with its nearest neighbor. One may sample pairs of disconnected nodes as negative examples, but such sampling method may produce false negatives, as disconnected nodes may or may not be semantically dissimilar.

variants of them (i.e., vanilla, denoising and variational). We leave the exhaustive comparison of the other variants for future work.

## 4.1 HANDWRITTEN DIGITS

The MNIST database is commonly used in the initial study of newly proposed methods due to its simplicity (LeCun et al., 1998). It contains $70,000$ images of handwritten digits (one digit per image); $10,000$ of these images are test data and the other $60,000$ are training data. The original task for the data set is multi-class classification. Since the proposed method is not a classifier but a representation learner (i.e., an encoder), we have evaluated our method using the following procedural: 1) we train the encoder with all the training data, 2) we encode both training data and test data into the learned representation space, 3) we train a simple classifier (i.e., linear support vector machine/SVM) with various amounts of labeled training data in the representation space, then apply the classifier to the representation of test data and report the classification error (i.e., semi-supervised classification problem), and 4) we also apply a clustering method (i.e., $k$-means) to the representation of test data and report the adjusted Rand index. As a proof of concept, we didn't put much effort in optimizing the structure of the encoder/encoder. We simply used a 4-layer $2D$ convolutional net as the encoder and a 4-layer transposed $2D$ convolutional net as the decoder. The detailed setting of the network architecture is summarized in Figure 4. We have tried several other convolutional net architectures as well; we draw the same conclusion from the experimental results with these alternative architectures.

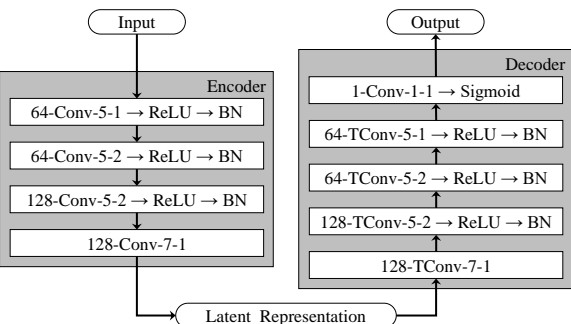

Figure 4: Network architecture for the decoder and decoder. 64-Conv-5-1 denotes $2D$ convolutional layer with 64 $5 \times 5$ kernels and stride of 1. ReLU denotes rectified linear unit. BN denotes batch normalization. TConv denotes transposed $2D$ convolutional layer.

Here we use the neighbor-encoder configuration (shown in Figure 2b) with the simple neighbor definition for our neighbor-encoder. We compare the performance of three variants (i.e., vanilla, denoising, and variational) of neighbor-encoder and the same three variants of autoencoder. Figure 5 shows the classification error rate as we change the number of labeled training data for linear SVM. All neighbor-encoder variants outperforms their corresponding autoencoder variants except the variational neighbor-encoder when the number of labeled training data is larger. Overall, denoising neighbor-encoder produce the most discriminating representations.

Besides the semi-supervised learning experiment, we also performed a clustering experiment with $k$-means which is purely unsupervised. Table 1 summarized the experiment result. The overall conclusion is similar to that of the semi-supervised learning experiment where all neighbor-encoder variants outperforms their corresponding autoencoder variants. Unlike the semi-supervised experiment, variational neighbor-encoder produces the most clusterable representations in this particular experiment, but all three variants of neighbor-encoder are comparable with each other.

Table 1: The clustering adjust Rand index with $k$-means.

|  | Vanilla | Denoising | Variational |
|---|---|---|---|
| Autoencocder | 0.3005 | 0.3710 | 0.4492 |
| Neighbor-encoder | 0.4926 | 0.5039 | 0.5179 |

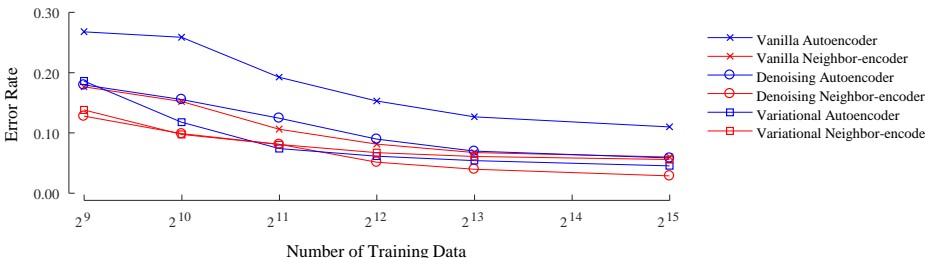

Figure 5: The classification error rate with linear SVM versus various training data size using different variants (i.e., vanilla, denoising, variational) of autoencoder and neighbor-encoder. Neighbor-encoder constantly outperforms autoencoder.

In the previous two experiments, we define the neighbor of an object as its 1st nearest neighbor under Euclidean distance. With such definition, the visual difference between an object with its neighbor using such definition is usually small given that we have sufficient data. To allow for more visual discrepancy between the objects and their neighbors, we could change that neighbor definition to the $i$th nearest neighbor under Euclidean distance ($i > 1$). We have repeated the clustering experiment under different setting of $i$ to examine the effect of increasing discrepancy between the objects and their neighbors. We chose to perform the clustering experiment instead of the semi-supervised learning experiment because 1) clustering is unsupervised and 2) it is easier to present the clustering result in a single figure as semi-supervised learning requires us varying both the number of training data and $i$.

Figure 6 summarizes the result, and Appendix A shows a randomly selected set of object-neighbor pair under different setting of $i$. The performance peaks around $i = 2^4$ and decreases as we increase $i$; therefore, choosing the $2^4$th nearest neighbor as the reconstruction target for neighbor-encoder would create enough discrepancy between the object-neighbor pair for better representation learning. When neighbor-encoder is used in this fashion, it can be regarded as a non-parametric way of generating noisy objects (similar as the principle of denoising autoencoder), and the settings of $i$ controls the amount of noise added to the object. Note that neighbor-encoder is not equivalent to denoising autoencoder as several objects can share the same $i$th nearest neighbor (recall Figure 3c), but denoising autoencoder would most likely generate different noisy inputs for different objects.

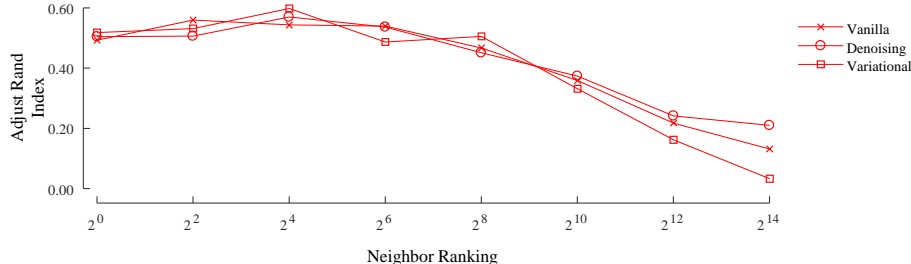

Figure 6: The clustering adjust Rand index versus the proximity of the neighbor using various neighbor-encoder variations (i.e., vanilla, denoising, variational). The proximity of a neighbor is defined as its ranking when query with the input. All three neighbor-encoder variations roughly reach their peak performance when the $2^4$ neighbor is used as the decoder target, and the performance declined afterward.

To explain the performance difference between autoencoder and neighbor-encoder, we randomly selected 5 test examples from each class (see Figure 7a) and fed them through both the autoencoder and the neighbor-encoder trained in the previous experiments. The outputs are shown in Figure 15 where the top row and bottom row are autoencoder (AE) and neighbor-encoder (NE) respectively. As expected, the output of autoencoder is almost identical to the input image. In contrast, although the output of neighbor-encoder is still very similar to the input image, the intra-class variation is reduced comparing the output of autoencoder. This is because neighbor-encoder tends to reconstruct the same neighbor image from similar input data points (recall Figure 3c). As a result, the

latent representation learned by neighbor-encoder is able to achieve better classification/clustering performance.

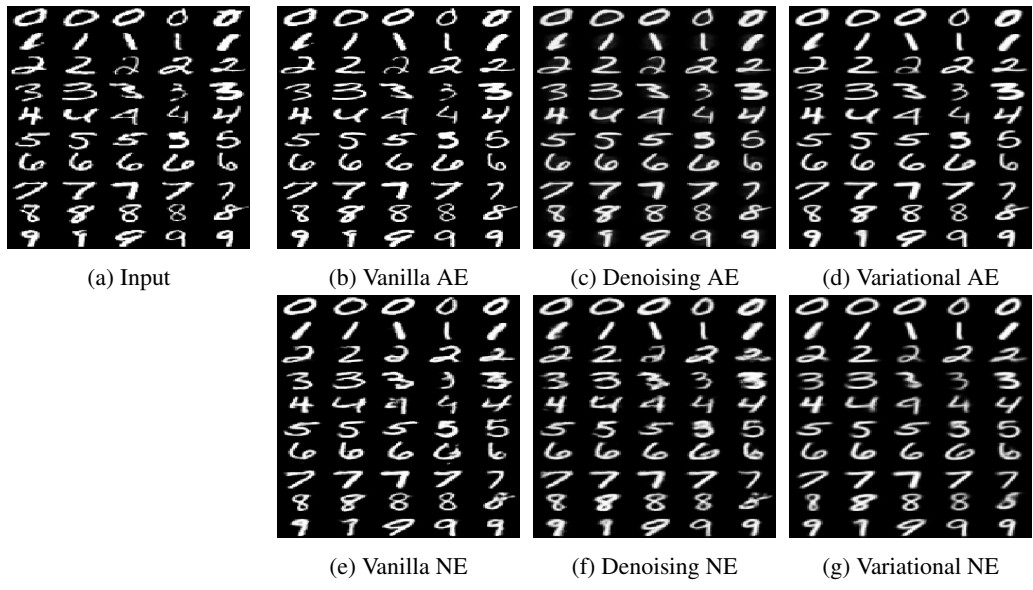

| (a) Input | (b) Vanilla AE | (c) Denoising AE | (d) Variational AE |
| --- | --- | --- | --- |

| (e) Vanilla NE | (f) Denoising NE | (g) Variational NE |
| --- | --- | --- |

Figure 7: Outputs of the decoders for different autoencoder (AE) and neighbor-encoder (NE) variations.

## 4.2 IMAGES

The CIFAR10 data set is collected for tiny image classification (Krizhevsky & Hinton, 2009). It contains $60,000$ $32 \times 32$ tiny colored images from $10$ different classes[3]; $10,000$ of these images are test data and the other $50,000$ are training data. We perform the experiment following the procedural outlined in Section 4.1 using the network architecture shown in Figure 8. We adopt a $4$-layer $2D$ convolutional net as the encoder and a $4$-layer transposed $2D$ convolutional net as the decoder similar to the last section, and we perform experiments with two different neighbor definitions: feature space neighbor and side information neighbor.

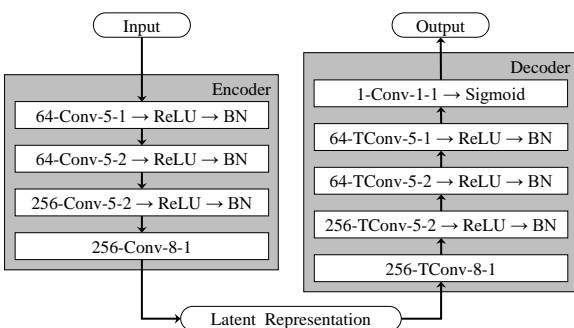

Figure 8: Network architecture for the decoder and decoder. 64-Conv-5-1 denotes $2D$ convolutional layer with $64$ $5 \times 5$ kernels and stride of $1$. ReLU denotes rectified linear unit. BN denotes batch normalization. TConv denotes transposed $2D$ convolutional layer.

Our first set of experiments are based on the feature space neighbor definition. The computer vision feature we used in this set of experiments is the standard bag-of-visual-words (BoVW) which follows the *dense SIFT → vector quantization*[4] pipeline. We use BoVW to define the neighbor of each

---

[3]airplane, automobile, bird, cat, deer, dog, frog horse, ship, and truck

[4]The codebook size is set to $1,024$

image, and raw pixel values to train autoencoder and neighbor-encoder. When the neighbor relationship is defined as such, 22% of the object-neighbor pairs are from the same class[5]. Figure 9 shows the result of the semi-supervised learning (classification) experiment comparing neighbor-encoder with autoencoder. Even though the provided object-neighbor pairs are noisy, neighbor-encoders still provides notably more discriminating representation comparing to their autoencoder counterparts when relatively few labeled training data are available; when labeled training data is abundant, neighbor-encoders are comparable to autoencoders. Overall, variational neighbor-encoder outperforms the other methods, especially when the amount of labeled training data is small.

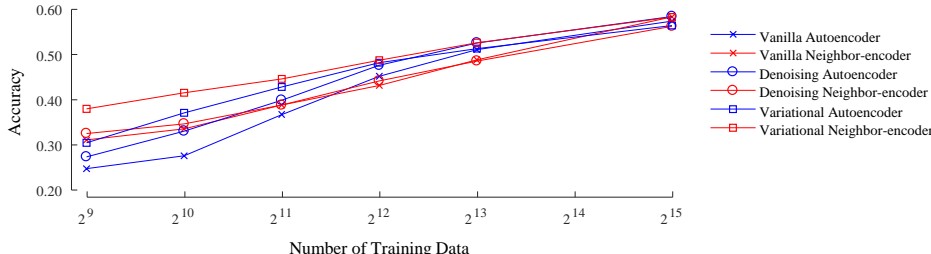

Figure 9: The classification accuracy with linear SVM versus various training data size using different variations (i.e., vanilla, denoising, variational) of autoencoder and neighbor-encoder (with feature space neighbor definition).

Next, Table 2 shows the result of $k$-means clustering experiment which aims to benchmark the performance of the learned representation in unsupervised learning tasks. The conclusion is similar to that of the semi-supervised experiment where the neighbor-encoders' adjust Rand index exceeds their autoencoder counterparts', but this time, the vanilla neighbor-encoder provides the most clusterable representation instead of variational neighbor-encoder. This result demonstrates that though the 1-nearest-neighbor information is relatively weak and noisy (only 22% of the object-neighbor pairs are from the same class), neighbor-encoder can still benefit from it.

Table 2: The clustering adjust Rand index with $k$-means. The neighbor-encoder representation is learned using feature space neighbor definition.

|                   | Vanilla | Denoising | Variational |
|-------------------|---------|-----------|-------------|
| Autoencocder      | 0.0535  | 0.0539    | 0.0424      |
| Neighbor-encoder  | 0.0609  | 0.0560    | 0.0535      |

Our second set of experiments are based on the side information neighbor definition. In order to test the performance of this neighbor definition, we synthetically created the side information for the CIFAR 10 data set by transforming the labels/classes of the original CIFAR 10 data to object-neighbor pairs. That is, for each object, we randomly assign another object from the same class as its neighbor. Note, with such transformation, we no longer have the dissimilarity information (i.e., which pairs of objects belong to different classes), nor do we know the number of classes in the data set. In other words, we cannot trivially apply normal supervised learning method on these side information neighbor pairs. Aside from examining another neighbor function, this set of experiments also examine the neighbor-encoder in a what-if scenario in which the 1-nearest neighbor information contains 100% accurate object-neighbor pairs inline with the evaluation task. To be fair with the autoencoder (as neighbor-encoder indirectly uses the label in this case), we modify the autoencoder to also use the neighbor information by adopting the contrastive loss (which is typically used in Siamese network as shown by Koch et al. (2015)) in addition to the self-reconstruction loss.

Figure 10 shows the result of the semi-supervised learning experiment (i.e. the classification experiment) with neighbor-encoder and autoencoder. First of all, a more accurate neighbor information improves the performance of neighbor-encoder, and the additional contrastive loss also boosts the performance of autoencoder (see Figure 2). By comparing the performance between the neighbor-encoder variants and the autoencoder variants, we reach a similar conclusion as the previous semi-

---

[5]If the neighbor of each object is randomly assigned, only 10% of the object-neighbor pairs belong to the same class.

supervised experiments: the three variants of neighbor-encoder notably outperform their autoencoder counterparts when small number of labeled training data is available, while the performance of neighbor-encoder is comparable to autoencoder when more labeled training data is made available to the linear SVM. Variation neighbor-encoder once again produces the most discriminating representation.

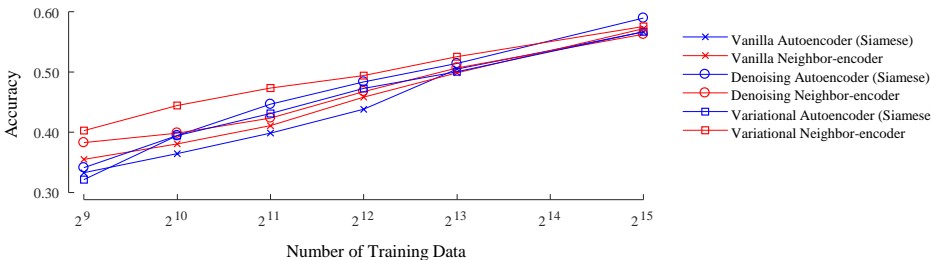

Figure 10: The classification accuracy with linear SVM versus various training data size using different variations (i.e., vanilla, denoising, variational) of either autoencoder or neighbor-encoder (with side information neighbor definition). The side information is added to the autoencoder by adding the contrastive loss to the objective function.

Table 3 summarizes the result of the clustering experiment. The performance of neighbor-encoder noticeably outperforms both the enhanced (with contrastive loss) autoencoder and the stock autoencoder (see Table 2) regardless of the variations (i.e., vanilla, denoising, variational), and the contrastive loss seems to hinder the clusterability of the representation produced by autoencoder. This time, denoising neighbor-encoder produces the representation with best clusterability, followed by vanilla neighbor-encoder which is the best neighbor-encoder for clustering using feature space neighbor definition. The fact that neighbor-encoder is able to achieve better performance with nearest neighbor reconstruction loss function comparing to contrastive loss function hints the possibility of adopting neighbor reconstruction loss function to one-shot/few-shot learning problem where the training data for each class is scarce.

Table 3: The clustering adjust Rand index with $k$-means. The neighbor-encoder representation is learned using side information neighbor definition. The side information is added to the autoencoder by adding the contrastive loss to the objective function.

|  | Vanilla | Denoising | Variational |
|---|---|---|---|
| Autoencocder (Siamese) | 0.0444 | 0.0402 | 0.0399 |
| Neighbor-encoder | 0.0934 | 0.1089 | 0.0710 |

## 4.3 HUMAN PHYSICAL ACTIVITIES

In Section 3, we have also introduced the $k$-neighbor-encoder in addition to the neighbor-encoder. Here we test the $k$-neighbor-encoder on the PAMAP2 data set (Reiss & Stricker, 2012a;b) using the time series subspace neighbor definition (Yeh et al., 2017). We choose the subspace neighbor definition because 1) it addresses one of the commonly seen multidimensional time series problem scenarios (i.e., the existence of irrelevant/noisy dimensions), 2) it is able to extract meaningful repeating patterns (Yeh et al., 2017), and 3) it naïvely gives multiple "types" neighbors to each object.

The PAMAP2 data set was collected by mounting 3 inertial measurement units and a heart rate monitor on 9 subjects while they were performing 18 different physical activities (e.g., walking, running, playing soccer) during 9 recording sessions ranging from 0.5 hours to 1.9 hours (i.e., 1 session per subject). The subjects performed one activity for few minutes, took a short break, then continued performing another activity. In order to transfer the data set into a format that we can use for evaluation (i.e., a training/test split), for each subject (or recording session), we cut the data into segments according to their corresponding physical activities; then, within each activity segment, we generate training data from the first half and test data from the second half with a sliding window of length of 100 and step size of 1. We make sure that there is no overlap between training data and test data. After the reorganization, we end up with 9 data sets (i.e., 1 pair of training/test set

per subject). We ran experiments on each data set independently and report averaged performance results.

The experiment procedural is very similar to the one presented in Section 4.1. We perform the experiments under two different scenarios: "clean" and "noisy". In the "clean" scenario, we manually deleted some dimensions of the data that are irrelevant (or harmful) to the classification/clustering tasks, while in the "noisy" scenario, all dimensions of the data are retained. The encoder-decoder network architecture we used is summarized in Figure 11. Here we use a 5-layer $1D$ convolutional net as the encoder and a 5-layer transposed $1D$ convolutional net as the decoder. Similar as in Section 4.1, we did not put much effort in optimizing the structure of this network architecture. We have tried modifying the convolutional net architectures in various ways, such as adding batch normalization, changing the number of layers, or varying the number of filters for each layer, etc., and the conclusion drawn from the experimental results remains virtually unchanged. During test time, we apply global averaging pooling to the output of the encoder to obtain the latent representation.

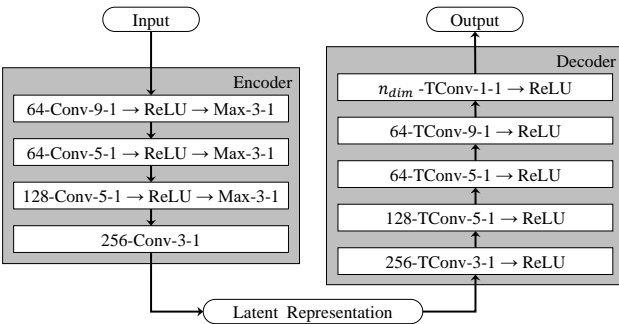

Figure 11: Network architecture for the encoder and the decoder. 64-Conv-9-1 denotes $1D$ convolutional layer with $64$ sized $9$ kernels and sized $1$ stride. ReLU denotes rectified linear unit. Max-3-1 denotes max pooling layer with sized $3$ pooling window and sized $1$ stride. TConv denotes transposed $1D$ convolutional layer. $n_{dim}$ is the number of dimension for the input multidimensional time series.

In Figure 12, we compare the semi-supervised classification capability of vanilla, denoising, variational autoencoder/$k$-neighbor-encoder under both the"clean" scenario and the "noisy" scenario. Both vanilla and denoising $k$-neighbor-encoder outperforms their corresponding autoencoder in all scenarios. The performance difference is more notable when the number of training data is small. On the contrary, variational autoencoder outperforms the corresponding $k$-neighbor-encoder; however, the performance of both variational autoencoder and $k$-neighbor-encoder are considerably worse comparing to their vanilla and denoising counterparts. Overall, both vanilla and denoising $k$-neighbor-encoder works relatively well for this problem.

Table 4 shows the clustering experiment with $k$-means. For vanilla encoder-decoder system, $k$-neighbor-encoder surpasses autoencoder in both scenarios, especially in the noisy scenario. When the denoising mechanism is added to the encoder-decoder system, it greatly boosts the performance of autoencoders, but the performance of $k$-neighbor-encoder is still overall superb comparing to autoencoder. Similar to the semi-supervised learning experiment, the variational encoder-decoder system performs poorly for this data set. In general, both vanilla and denoising $k$-neighbor-encoder outperforms the their autoencoder counterparts for the clustering problem on PAMAP2 data set.

Table 4: The clustering adjust Rand index with $k$-means. Both vanilla and denoising $k$-neighbor-encoder outperforms their autoencoder counterparts while all the variational variants performs poorly.

|  |  | Vanilla | Denoising | Variational |
|---|---|---|---|---|
| Clean | Autoencocder | 0.3815 | 0.4159 | 0.1597 |
|  | Neighbor-encoder | 0.4203 | 0.4272 | 0.1192 |
| Noisy | Autoencocder | 0.0890 | 0.1763 | 0.0825 |
|  | Neighbor-encoder | 0.1844 | 0.1817 | 0.1111 |

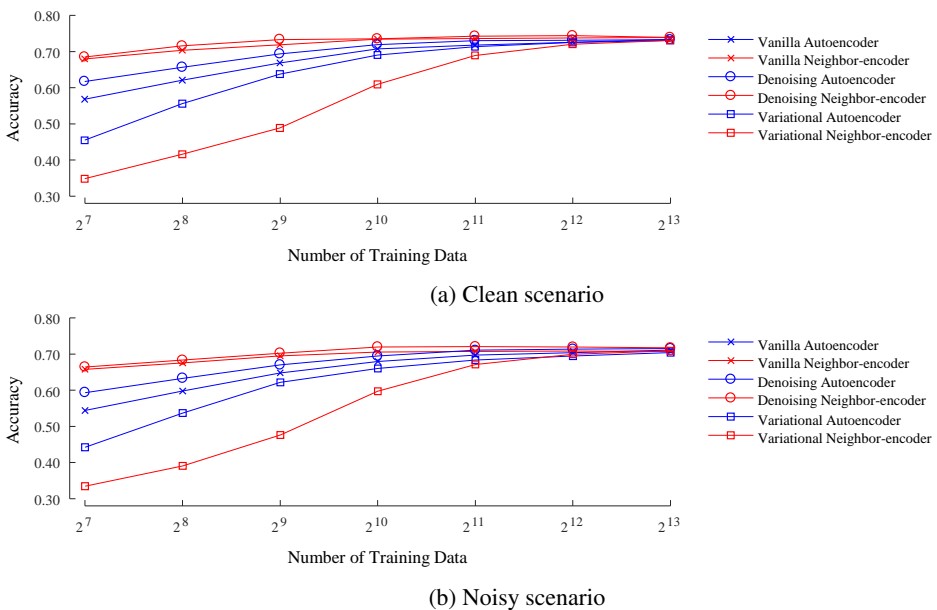

(a) Clean scenario

(b) Noisy scenario

Figure 12: The classification accuracy with linear SVM versus various labeled training data size using different variants (i.e., vanilla, denoising, variational) of either autoencoder and $k$-neighbor-encoder. Both vanilla neighbor-encoder and denoising neighbor-encoder outperform their corresponding autoencoder while both variational neighbor-encoder and variational autoencoder perform poorly when number of labeled training data is small.

Figure 1 further demonstrates the advantage of neighbor-encoder over autoencoder. Here we use $t$-SNE to project various representations of the data of subject $1$ into $2D$ space. The representations include the raw data itself, the latent representation learned by denoising autoencoder, and the latent representation learned by denoising $k$-neighbor-encoder. Despite the clustering experiment suggest us autoencoder is comparable with $k$-neighbor-encoder, we can see that the latent representation learned by $k$-neighbor-encoder provides a much more meaningful visualization of different classes than the rival methods (includes autoencoder) in the face of noisy/irrelevant dimensions.

## 4.4 INSTRUMENTAL SOUNDS

To further demonstrate the versatility of the neighbor-encoder framework, we conducted experiments on a data set for predominant instrumental sound recognition in polyphonic music, using a benchmark data set collected by Fuhrmann (2012). The training data consists of $6,951$ 3-second long music clips, with each clip labeled with one of the following primary instruments: cello, clarinet, flute, acoustic guitar, electric guitar, Hammond organ, piano, saxophone, trumpet, violin, or singing voice. We performed semi-supervised classification and clustering experiments by applying 10-fold cross validation on the training data. Similar to Section 4.1, we use accuracy and adjusted Rand index as the performance metric for semi-supervised classification and clustering respectively.

The neighborhood function we adopted for this data set is the feature space neighbor definition. Assume that we have the MFCC of an audio clip (which has 20 coefficients and 130 time frames); we can then evaluate its mean $\mu$ and standard deviation $\sigma$ across time, and use the resulting 40-dimensional vector (which is produced by concatenating $\mu$ and $\sigma$) to represent the audio clip. We define the nearest neighbors of this audio clip as the clips closest to it in this 40-dimensional space. With such neighbor definition, $49\%$ of the object-neighbor pairs belong to the same class (average across 10 folds)[6]. Note that this 40-dimensional feature vector is only used in the neighborhood function; after the nearest neighbors of an audio clip is found, we use the mel-frequency spectrum of the audio clip and its neighbors as the input/output features of the encoder-decoder network.

---

[6]If the neighbor of each object is randomly assigned, only $9\%$ of the object-neighbor pairs belong to the same class.

We use *librosa* to extract both MFCC and mel-frequency spectrum of the audio clips (McFee et al., 2015). The only non-default setting we use here is the windows size for short-time Fourier transform. The default value is $2,048$, and we use $1,024$. Under our feature extraction settings, the MFCC has 20 coefficients and 130 time frames and mel-frequency spectrum has 128 mel-frequency bins and 130 time frames.

We use a 5-layer $2D$ convolutional net as the encoder and a 5-layer transposed $2D$ convolutional net as the decoder. The encoder-decoder network architecture is summarized in Figure 13. In each iteration of stochastic gradient descent, we randomly selected a mini-batch of music clips, and within each clip, we randomly selected a segment of 16 consecutive frames. In other words, the network is trained on segments within the music clips instead of the music clip as a whole. The same sampling process is also applied to their corresponding neighbor, but instead of selecting randomly, we select the segment that most similar to the input segment. The full clip is used at the test phase, and we apply global averaging pooling to the output of the encoder to obtain the latent representation. We have tried several other network architecture and training schemes (e.g., varying the length of segment), and the conclusions drawn from the experimental result are essentially the same.

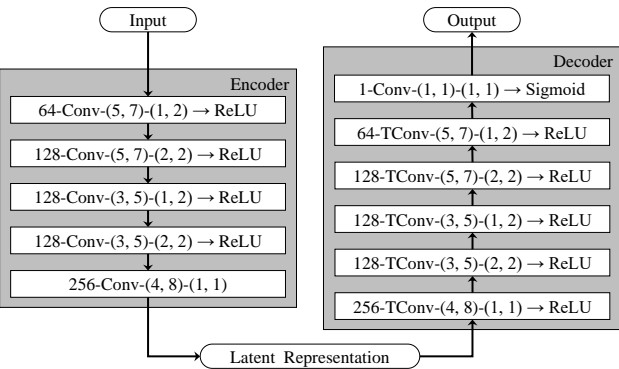

Figure 13: Network architecture for the decoder and decoder. 64-Conv-$(5, 7)$-$(1, 2)$ denotes a $2D$ convolutional layer with 64 $5 \times 7$ kernels and stride of $(1, 2)$. ReLU denotes rectified linear unit. TConv denotes transposed $2D$ convolutional layer. The first dimension within the parenthesis is time, and the second dimension is frequency.

The results of the semi-supervised learning experiment with various autoencoder and neighbor-encoder are shown in Figure 14. Overall, vanilla neighbor-encoder almost always achieves the highest classification accuracy. Unlike previous experiment result, the gap between neighbor-encoder and autoencoder does not reduce as the number of training data increase.

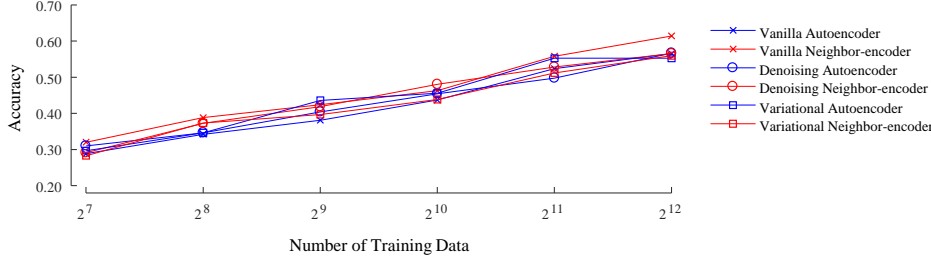

Figure 14: The classification accuracy with linear SVM versus various training data size using different variations (i.e., vanilla, denoising, variational) of autoencoder and neighbor-encoder.

Table 4 summarizes the $k$-means clustering experiment result. The method that achieves the highest adjust Rand index is variational autoencoder, followed by vanilla neighbor-encoder. Similar to the semi-supervised learning result (i.e., first point in Figure 14), the best neighbor-encoder (i.e., vanilla neighbor-encoder) is comparable to the best autoencoder (i.e., variational autoencoder) when the entire pipeline uses no labels.

Table 5: The clustering adjust Rand index with $k$-means.

|  | Vanilla | Denoising | Variational |
|---|---|---|---|
| Autoencocder | 0.0537 | 0.0519 | 0.0688 |
| Neighbor-encoder | 0.0640 | 0.0581 | 0.0621 |

## 5 CONCLUSION

In this work, we have proposed an unsupervised learning framework called neighbor-encoder that is both *general*, in that it can easily be applied to data in various domains, and *versatile* as it can incorporate domain knowledge by utilizing different neighborhood functions. We have showcased the effectiveness of neighbor-encoder compared to autoencoder in various domains, including images, time series, music, etc. In future work, we plan to either 1) explore the possibility of apply neighbor-encoder to problems like one-shot learning or 2) demonstrate the usefulness of the neighbor-encoder in more practical and applied tasks, including information retrieval.

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

# A NEIGHBOR PAIRS FOR HANDWRITTEN DIGITS

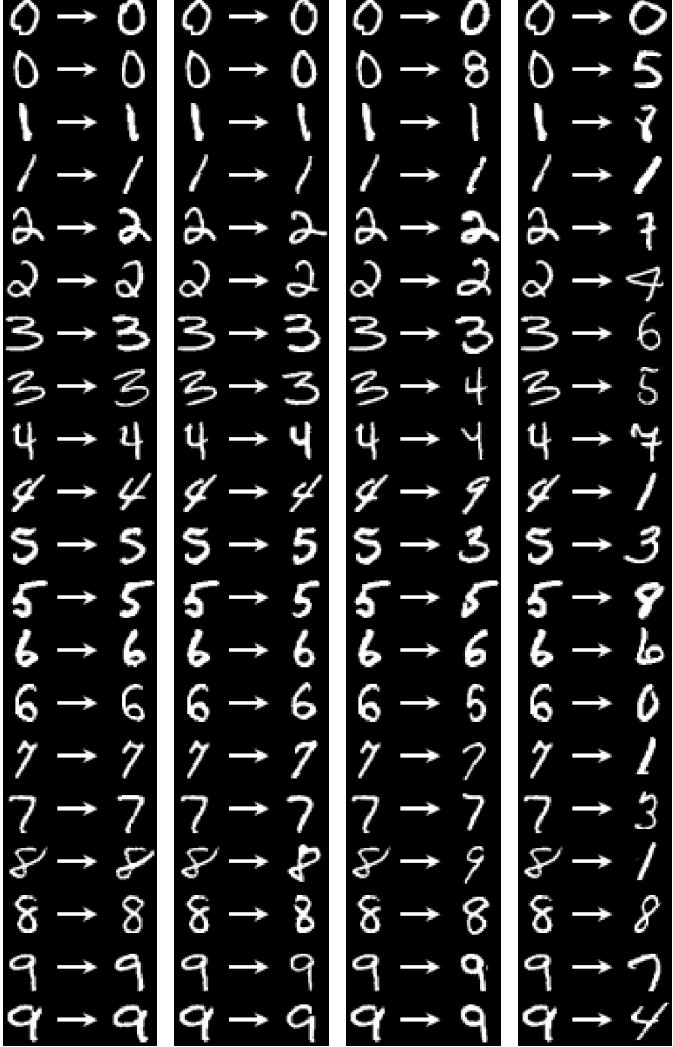

(a) $2^0$ neighbor   (b) $2^4$ neighbor   (c) $2^8$ neighbor   (d) $2^{12}$ neighbor

Figure 15: Neighbor pairs under different proximity setting.

