# OpenReview forum: "Neighbor-encoder"
_ICLR.cc/2018/Conference — Reject_

### Official Review · AnonReviewer1 · 2017-11-22
**Lack of comparison with other autoencoders**

**Rating:** 5
**Confidence:** 5

**Review:**

This paper describes a generalization of autoencoders that are trained to reconstruct a close neighbor of its input, instead of merely the input itself. Experiments on 3 datasets show that this yields better representations in terms of post hoc classification with a linear classifier or clustering, compared to a regular autoencoder.

As the authors recognize, there is a long history of research on variants of autoencoders. Unfortunately this paper compares with none of them. While the authors suggest that, since these variations can be combined with the proposed neighbor reconstruction variant, it's not necessary to compare with these other variations, I disagree. It could very well be that this neighbor trick makes other methods worse for instance.

At the very least, I would expect a comparison with denoising autoencoders, since they are similar if one thinks of the use of neighbors as a structured form of noise added to the input. It could very well be in fact that simply adding noise to the input is sufficient to force the autoencoder to learn a valuable representation, and that the neighbor reconstruction approach is simply an overly complicated approach of achieving the same results. This is an open question right now that I'd expect this paper to answer.

Finally, I think results would be more impressive and likely to have impact if the authors used datasets that are more commonly used for representation learning, so that a direct performance comparison can be made with previously published results. CIFAR 10 and SVHN would be good alternatives.

Overall, I'm afraid I must recommend that this paper be rejected.

---

> ### Author Response · Authors · 2018-01-05
> **We really appreciate your valuable review**
>
> Dear Reviewer,
>
> We really appreciate your valuable review! We have modified our paper based on your feedback by:
>
> 1) adding denoising and variational autoencoder (and their neighbor-encoder counterparts) to all experiments, and …
> 2) adding a new set of experiment on CIFAR 10 in Section 4.2. In all the experiments, we observed that neighbor-encoder and its variants outperform their autoencoder counterparts when applied in semi-supervised classification (when the number of labeled data available is small) and clustering tasks.
>
> Thanks,
> Authors

---

### Official Review · AnonReviewer3 · 2017-11-27
**Nice Idea but what about "Curse of Dimensionality"?**

**Rating:** 6
**Confidence:** 4

**Review:**

A representation learning framework from unsupervised data, based not on auto-encoding (x in, x out), but on neighbor-encoding (x in, N(x) out, where N(.) denotes the neighbor(s) of x) is introduced.

The underlying idea is interesting, as such, each and every degree of freedom do not synthesize itself similar to the auto-encoder setting, but rather synthesize a neighbor, or k-neighbors. The authors argue that this form of unsupervised learning is more powerful compared to the standard auto-encoder setting, and some preliminary experimental proof is also provided.

However, I would argue that this is not a completely abstract - unsupervised representation learning setting since defining what is "a neighbor" and what is "not a neighbor" requires quite a bit of domain knowledge. As we all know, the euclidian distance, or any other comparable norm, suffers from the "Curse of Dimensionality" as the #-of-Dimensions increase.

For instance, in section 4.3, the 40-dimensional feature vector space is used to define neighbors in. It would be great how the neighborhood topology in that space looks like.

All in all, I do like the idea as a concept but I am wary about its applicability to real data where defining a good neighborhood metric might be a major challenge of its own.

---

> ### Author Response · Authors · 2018-01-05
> **Thank you for your helpful review and kind words**
>
> Dear Reviewer,
>
> Thank you for your helpful review and kind words! We are glad that you like the idea.
>
> In the review, you have argued that the neighbor-encoder method is not a completely abstract-unsupervised representation learning method as it requires domain knowledge to define the neighbor relationship. This statement is certainly valid, as we do need some domain knowledge. However, the amount of domain knowledge required by neighbor-encoder is minimal in comparison to what is required by a typical supervised representation learning method: we only need a "neighbor" to be defined, the "non-neighbor" information is not needed. In other words, we only need to know what is "similar" (and this information can be very sparse), but not what is "not similar" (the key information needed to divide objects into different classes/clusters).
> Furthermore, note that the domain knowledge provided do not need to be precise. Our MINST example in Section 4.1 simply use Euclidean distance in raw pixel space as the similarity measure to find the neighbors. For the newly added CIFAR10 data set Section 4.2, we use Euclidean distance in a common computer vision feature space as the similarity measure; the feature selected does not have much discriminative power for this data set and only 22% of the object-neighbor pairs are from the same class. Nevertheless, the results (Figure 9 and Table 2) show that all three variants of neighbor-encoder outperform their autoencoder counterparts in both semi-supervised classification (when number of labeled data is small) and clustering tasks.
>
> To clarify, our claim is not that neighbor-encoder is a purely unsupervised representation learning method. Instead, our claim is that even a tiny amount of domain knowledge can greatly improve unsupervised representation learning, and neighbor-encoder is an effective way to incorporate such domain knowledge into the unsupervised representation learning framework.
>
> For any comparable norm based neighbor definition, "curse of dimensionality" indeed would be a problem. To quantify the severity of such problem, we measured the percentage of object-neighbor pairs being in the same class. For example, in Section 4.4 (originally Section 4.3), about 49% of the object-neighbor pairs in the 40-dimensional feature vector space are in the same class (note that this is relatively high, as the default rate for randomly assigned neighbor is just ~9% for this data set). Another way we envision that can further increase this percentage is to use side information to define a neighbor (as introduced in Section 3.4). For instance, images/document on the same webpage or reviews of the same paper/movie/music could be declared being neighbor of each other. Such side information would much less sensitive to the curse of dimensionality.
>
> Thanks,
> Authors

---

### Official Review · AnonReviewer2 · 2017-11-28
**Review: neighbor-encoder -> neighbor encoder**

**Rating:** 4
**Confidence:** 4

**Review:**

This paper presents a variant of auto-encoder that relaxes the decoder targets to be neighbors of a data point. Different from original auto-encoder, where data point x and the decoder output \hat{x} are forced to be close, the neighbor-encoder encourage the decoder output to be similar to the neighbors of the input data point. By considering the neighbor information, the decoder targets would have smaller intra-class distances, thus larger inter-class distances, which helps to learn better separated latent representation of data in terms of data clusters. The authors conduct experiments on several real but relative small-scale data sets, and demonstrate the improvements of learned latent representations by using neighbors as targets.

The method of neighbor prediction is a simple and small modification of the original auto-encoder, but seems to provide a way to augment the targets such that intra-class distance of decoder targets can be tightened. Improvements in the conducted experiments seem significant compared to the most basic auto-encoder.

Major issues:

There are some unaddressed theoretical questions. The optimal solution to predict the set of neighbor points in mean-squared metric is to predict the average of those points, which is not well justified as the averaged image can easily fall off the data manifold. This may lead to a more blurry reconstruction when k increases, despite the intra-class targets are tight. It can also in turn harm the latent representation when euclidean neighbors are not actually similar (e.g. images in cifar10/imagenet that are not as simple as 10 digits). This seems to be a defect of the neighbor-encoder method and is not discussed in the paper.

The data sets used in the experiments  are relatively small and simple, larger-scale experiments should be conducted. The fluctuations in Figure 9 and 10 suggest the significant variances in the results. Also, more complicated data/images can decrease the actual similarities of euclidean neighbors, thus affecting the results.

The baselines are weak. Only the most basic auto-encoder is compared, no additional variants or other data augmentation techniques are compared. It is possible other variants improve the basic auto-encoder in similar ways.

Some results are not very well explained. It seems the performance increases monotonically as the number of neighbors increases (Figure 5, 9, 10). Will this continue or when will the performance decrease? I would expect it to decrease as the far away neighbors will be dissimilar. The authors can either attach the nearest neighbors figures or their statistics, and provide explanations on when and why the performance decrease is expected.

Some notations are confusing and need to be improved. For example, X and Y are actually the same set of images, the separation is a bit confusing; y_i \in y in last paragraph of page 4 is incorrect, should use something like y_i in N(y).

---

> ### Author Response · Authors · 2018-01-05
> **Thank you very much for your valuable review**
>
> Dear Reviewer,
>
> Thank you very much for your valuable review. Addressing your concerns has made our paper much stronger. Our responses to the major issues are listed below:
>
> Issue 1: There are some unaddressed theoretical questions. The optimal solution to predict the set of neighbor points in mean-squared metric is to predict the average of those points, which is not well justified as the averaged image can easily fall off the data manifold. This may lead to a more blurry reconstruction when k increases, despite the intra-class targets are tight. It can also in turn harm the latent representation when euclidean neighbors are not actually similar (e.g. images in cifar10/imagenet that are not as simple as 10 digits). This seems to be a defect of the neighbor-encoder method and is not discussed in the paper.
> Response to Issue 1: Thank you for raising this concern. The issue is addressed by removing the original configuration in question (in which we randomly selected one of the k nearest neighbors as the target to predict) as it does have this "averaging" problem. All the experiments are rerun with the most basic neighbor-encoder setting, in which we predict only the nearest neighbor of each object. As the target to predict is fixed, we no longer suffer from the "averaging neighbors" problem.
>
> Issue 2: The data sets used in the experiments are relatively small and simple, larger-scale experiments should be conducted. The fluctuations in Figure 9 and 10 suggest the significant variances in the results. Also, more complicated data/images can decrease the actual similarities of euclidean neighbors, thus affecting the results.
> Response to Issue 2: After we rerun all the experiments described in response to Issue 1, we no longer see significant variance in the results. A new set of experiment on CIFAR 10 is performed and reported in Section 4.2. We also included experiments comparing three variants of neighbor-encoder (vanilla, denoising, variational) with their autoencoder counterparts.
>
> Issue 3: The baselines are weak. Only the most basic auto-encoder is compared, no additional variants or other data augmentation techniques are compared. It is possible other variants improve the basic auto-encoder in similar ways.
> Response to Issue 3: We added comparison to two more popular variants of autoencoder, the denoising and variational autoencoder, in all of our experiments.
>
> Issue 4: Some results are not very well explained. It seems the performance increases monotonically as the number of neighbors increases (Figure 5, 9, 10). Will this continue or when will the performance decrease? I would expect it to decrease as the far away neighbors will be dissimilar. The authors can either attach the nearest neighbors figures or their statistics, and provide explanations on when and why the performance decrease is expected.
> Response to Issue 4: We believe that Figure 6 addresses this issue. A new set of experiments is performed by using neighbors that are further away (i.e., changing 1st neighbor to the ith nearest neighbor). The performance decreases as expected when i is larger than 16 because the performance is crippled by lower quality neighbors. Figure 15 shows example neighbor pairs under different proximity settings.
>
> Issue 5: Some notations are confusing and need to be improved. For example, X and Y are actually the same set of images, the separation is a bit confusing; y_i \in y in last paragraph of page 4 is incorrect, should use something like y_i in N(y).
> Response to Issue 5: The notation is improved as suggested.
>
> Thanks,
> Authors

---

### Public Comment · (anonymous) · 2018-03-05
**Replication of results / Loss Convergence**

I am trying to replicate your results with MNIST. Just to confirm (before I jump to conclusions or present my findings).
Would it be fair to say that just changing the optimisation function to reconstruct the neighbours as well as the input with a simple metric like MSE would be suffice (instead of seperate decoders)?

From what I gather, the paper also suggests that the architecture is more powerful in presence of noise (in comparison to existing AE architectures?

---

> ### Author Response · Authors · 2018-03-06
> **Thank you for your interest in our work**
>
> Q: Would it be fair to say that just changing the optimization function to reconstruct the neighbors as well as the input with a simple metric like MSE would be suffice (instead of separate decoders)?
> A: First, thanks for your interest. Do you mean that training the decoder to output a 28 x 56 image (containing both the input’s reconstruction and the neighbor’s reconstruction)?
> In the MNIST experiment we presented in the paper, the output of the decoder is always a 28 x 28 image containing either the input (in the case of autoencoder) or a neighbor (in the case of neighbor-encoder).
>
> Q: From what I gather, the paper also suggests that the architecture is more powerful in presence of noise (in comparison to existing AE architectures?
> A: It is an observation we made when comparing AE versus NE on the human physical activities data set as we are using a neighbor mining technique which ignores noisy dimensions.
> Our suggestion is that training by neighbor reconstruction instead of self-reconstruction yields better representation for semi-supervised classification and clustering. Since the proposed method is just changing the reconstruction target of existing AE architectures, it can be applied to most existing AE architectures. Based on our experimental result (with vanilla, denosing, and variational architectures), different architecture excels on different data set. Our main finding is on the effect of changing the reconstruction target (neighbor versus self) rather than the architectures.

---

### Decision · Program_Chairs · 2018-01-29
**ICLR 2018 Conference Acceptance Decision**

**Decision:**

Reject

**Comment:**

The paper proposes a form of autoencoder that learns to predict the neighbors of a given input vector rather than the input itself.  The idea is nice but there are some reviewer concerns about insufficient evaluation and the effect of the curse of dimensionality.  The revised paper does address some questions and includes additional helpful experiments with different types of autoencoders.  However, the work is still a bit preliminary.  The area of auto-encoder variants, and corresponding experiments on CIFAR-10 and the like, is crowded.  In order to convince the reader that a new approach makes a real contribution, it should have very thorough experiments.  Suggestions:  try to improve the CIFAR-10 numbers (they need not be state-of-the-art but should be more credible), adding more data sets (especially high-dimensional ones), and analyzing the effects of factors that are likely to be important (e.g. dimensionality, choice of distance function for neighbor search).